# A Reduction for Efficient LDA Topic Reconstruction

**Matteo Almanza**[*]
Sapienza University
Rome, Italy
almanza@di.uniroma1.it

**Flavio Chierichetti**[†]
Sapienza University
Rome, Italy
flavio@di.uniroma1.it

**Alessandro Panconesi**[‡]
Sapienza University
Rome, Italy
ale@di.uniroma1.it

**Andrea Vattani**
Spiketrap
San Francisco, CA, USA
avattani@cs.ucsd.edu

## Abstract

We present a novel approach for LDA (Latent Dirichlet Allocation) topic reconstruction. The main technical idea is to show that the distribution over the documents generated by LDA can be transformed into a distribution for a much simpler generative model in which documents are generated from *the same set of topics* but have a much simpler structure: documents are single topic and topics are chosen uniformly at random. Furthermore, this reduction is approximation preserving, in the sense that approximate distributions — the only ones we can hope to compute in practice — are mapped into approximate distribution in the simplified world. This opens up the possibility of efficiently reconstructing LDA topics in a roundabout way. Compute an approximate document distribution from the given corpus, transform it into an approximate distribution for the single-topic world, and run a reconstruction algorithm in the uniform, single-topic world — a much simpler task than direct LDA reconstruction. We show the viability of the approach by giving very simple algorithms for a generalization of two notable cases that have been studied in the literature, $p$-separability and matrix-like topics.

## 1   Introduction

Latent Dirichlet Allocation (henceforth LDA) is a well-known paradigm for topic reconstruction (Blei *et al.*, 2003). The general goal of topic reconstruction is, given a corpus of documents, to reconstruct the topics. LDA is a generative model according to which documents are generated from a given set of unknown topics, where each topic is modelled as a probability distribution over the words. One of the main motivations behind LDA is to allow documents to be able to talk about about multiple topics, a goal achieved by the following mechanism. To generate a document containing $\ell$ words we first select a probability distribution, the so-called *admixture*, over the topics. The admixture is randomly drawn from a Dirichlet distribution, hence the name. Then, the words of the document are selected one after the other in sequence by first selecting a topic at random according to the admixture, and then by randomly selecting a word according to the selected topic (which, as remarked, is just a

---

[*]Supported in part by the ERC Starting Grant DMAP 680153, and by the "Dipartimenti di Eccellenza 2018-2022" grant awarded to the Dipartimento di Informatica at Sapienza.

[†]Supported in part by the ERC Starting Grant DMAP 680153, by a Google Focused Research Award, and by the "Dipartimenti di Eccellenza 2018-2022" grant awarded to the Dipartimento di Informatica at Sapienza.

[‡]Supported in part by the ERC Starting Grant DMAP 680153, by a Google Focused Research Award, by the "Dipartimenti di Eccellenza 2018-2022" grant awarded to the Dipartimento di Informatica at Sapienza, and by BiCi – Bertinoro international Center for informatics.

probability distribution over the words). In this way all topics contribute to generate a document– to a degree specified for each document by a random admixture. To generate another document, another admixture is selected at random, and the same process is repeated. And so on, so forth.

In this paper we are interested in the problem of LDA topic identifiability which, roughly speaking, can be stated as follows: given a corpus of documents generated by the mechanism just described, reconstruct as efficiently and accurately as possible the $K$ unknown topics (in the paper $K$ will always denote the number of topics). LDA is actually more general than a mechanism for generating corpora of text documents, but it helps the intuition to consider it as a generative framework for text documents and we will stick to this scenario.

This paradigm has attracted a lot of interest, e.g., (Hong & Davison, 2010; Weng *et al.* , 2010; Zhao *et al.* , 2011; Yan *et al.* , 2013; Sridhar, 2015; Alvarez-Melis & Saveski, 2016; Li *et al.* , 2016; Hajjem & Latiri, 2017). Several algorithms for LDA topic reconstruction have been proposed (see, for instance, (Arora *et al.* , 2012, 2013; Anandkumar *et al.* , 2013; Bansal *et al.* , 2014)). In this paper we continue this line of research by presenting a novel approach, the main thrust of which is, loosely speaking, that of reducing the problem of topic identifiability in the LDA framework to the problem of topic identifiability under a much more constrained and simpler generative model. The simplified generative mechanism we have in mind is the following. The admixture, instead of being randomly selected anew for each document from a Dirichlet distribution, will stay put: when generating a document, a topic is selected uniformly at random with probability $1/K$. The second feature of the simplified framework is that documents are *single topic*, *i.e.* once a topic is selected, all the words in the document are chosen according to the distribution specified by that topic. We shall refer to this mechanism as *single topic allocation*, denoted as STA. In a nutshell, the contribution of this paper is to show that if we have an efficient and accurate algorithm for STA topic identifiability — a task seemingly much less daunting than its LDA counterpart — we can use it for efficient and accurate reconstruction of topics under the LDA paradigm. More precisely, we can do this in the case of *uniform* LDA, *i.e.* when the admixtures come from a symmetric Dirichlet distribution with a given parameter $\alpha$, which is a very important and commonly adopted special case (Blei *et al.* , 2003). Historically, STA-type models have been considered before the advent of LDA (see, e.g., (Nigam *et al.* , 2000)), whose main motivation, as mentioned, was precisely that of allowing documents to be mixtures of topics. In a way, our result vindicates STA in the sense that it shows that LDA reconstruction is not more general than STA reconstruction.

The main technical tool to achieve this is a reduction between the two paradigms, STA and uniform LDA. Given a set $\mathcal{T}$ of $K$ topics and a Dirichlet parameter $\alpha$, let $\mathcal{D} = \mathcal{D}_\ell$ be the distribution that they induce via LDA over the documents of a given length $\ell$. Similarly, let $\mathcal{S} = \mathcal{S}_\ell$ denote the distribution induced by STA over the documents of the same length $\ell$ *when the same set of topics $\mathcal{T}$ is used*. In a companion paper (Chierichetti *et al.* , 2018), we show that there is a reduction such that $\mathcal{S}$ can be computed from $\mathcal{D}$ and $\alpha$, and viceversa. In that paper this fact is used to derive impossibility results about LDA topic reconstruction whose gist is the following: unless the length of the documents is greater than or equal to the number of topics, identifying them is impossible. Here, we show how to exploit this reduction in the opposite direction: if we have an efficient algorithm for identifying the topics under STA then, thanks to the reduction, we can also use it to identify them under LDA.

Note that the above reduction deals with the exact probability distributions $\mathcal{D}_\ell$ and $\mathcal{S}_\ell$ over the documents, something which is helpful when impossibility results are concerned, but that becomes an issue if we are seeking reconstruction algorithms that have to be deployed in practice, and which have a limited number of documents to analyze. A first contribution of this paper is to show a robust version of the above reduction. Fix a set of topics $\mathcal{T}$, and suppose to have an approximation $\widetilde{\mathcal{D}}_\ell$ of the true distribution $\mathcal{D}_\ell$ induced by LDA when $\mathcal{T}$ is the set of topics. In practice, $\widetilde{\mathcal{D}}_\ell$ can be obtained from a large enough corpus of documents in a rather straightforward manner. Suppose also, as it is customarily assumed in practice, to know the value of the Dirichlet parameter $\alpha$. The robust version of the reduction, on input $\widetilde{\mathcal{D}}_\ell$ and $\alpha$, produces a distribution $\widetilde{\mathcal{S}}_\ell$ which is a good approximation of $\mathcal{S}_\ell$, the true distribution induced by STA when $\mathcal{T}$ is the set of topics.

This result suggests an intriguing possibility, namely that LDA topics could be identified in a rather roundabout way by means of the following pipeline. Starting from a document corpus generated by LDA from a set of hidden topics $\mathcal{T}$ that we wish to reconstruct, compute $\widetilde{\mathcal{D}}$, an approximation of the true document distribution $\mathcal{D}$. Apply the robust version of the reduction to $\widetilde{\mathcal{D}}$ and $\alpha$ (the Dirichlet

parameter which, as remarked, is assumed to be known in practice) to obtain $\widetilde{\mathcal{S}}$, an approximation of the true distribution $\mathcal{S}$ induced by STA from the same set of topics $\mathcal{T}$. Suppose now to have an efficient algorithm that, given $\widetilde{\mathcal{S}}$, outputs $\mathcal{T}'$, a good approximation of the set $\mathcal{T}$ of the unknown topics we are looking for. With such an algorithm we can solve LDA identifiability via single-topic distributions!

An algorithm capable of producing such a good approximation $\mathcal{T}'$ from $\widetilde{\mathcal{S}}$ is called *robust* in this paper. As hinted at by the above discussion, the second contribution of this paper is to show that the pipeline just described can be made to work. We provide a robust algorithm with provable guarantees with which we can solve in one stroke a natural generalization of two notable cases that have been studied in the literature. The first concerns so-called *separable* topics (Arora *et al.* , 2012, 2013). A set of topics is $p$-separable if, for each topic $T$ there is a word $w$ such that $T$ assigns probability at least $p$ to $w$ and every other topic assigns it probability zero. These special words are called *anchor words*. Thus, separability occurs when each topic is essentially identified uniquely by its anchor word. This set up has received considerable attention and several algorithms for LDA reconstruction have been proposed. One of the virtues of the $p$-separability assumption is that it makes it possible to derive algorithms with provable guarantees. For instance, the main result of Arora *et al.* (2012) states that there is an algorithm such that if a set of LDA topics are $p$-separable then they are identifiable within additive error $\delta$ in the $\ell_\infty$-norm, provided that the corpus contains

$$\Theta\left(\frac{K^6}{\delta^2 p^6 \gamma^2 \ell} \cdot \log m\right) \tag{1}$$

many documents, or more. In the expression, $m$ is the size of the vocabulary, $\ell$ is the length of the documents and $\gamma$ is the condition number of the topic-topic covariance matrix. As remarked by the same authors however, this algorithm is computationally impractical. A follow-up paper shows how to mitigate the problem by implementing the main steps in a different way (Arora *et al.* , 2013). The resulting algorithm is much more efficient but, unfortunately, heuristic in nature, thus losing one of the nice features of its computationally more expensive predecessor.

The second scenario we tackle is that of Griffiths & Steyvers (2004) in which Gibbs sampling is proposed as a heuristic without any performance guarantees for LDA topic reconstruction. In that paper, Gibbs sampling is applied to a dataset whose underlying set of topics is assumed to have the following structure. The vocabulary consists of a $n \times n$ matrix — each entry is a word (the authors of Griffiths & Steyvers (2004) consider $5 \times 5$ matrices, *i.e.* 25 words in total). There are $2n$ topics, each corresponding to a row or a column of the matrix. The topic corresponding to a given row has all zero entries except for that row, whose entries are uniformly $1/n$. Topics corresponding to columns are defined analogously. Note that this set of topics is not $p$-separable, since every word has positive probability in at least two topics (its row, and its column).

Both scenarios can be captured at once with the following natural definition. A set $\mathcal{T}$ of topics is $(p, t)$-separable if, for every topic $T \in \mathcal{T}$, there is a set of words $S_T$ of $t$ words such that *(i)* the product of the probabilities assigned by $T$ to the words of $S_T$ is at least $p$, and moreover *(ii)* for every other topic $T' \in \mathcal{T} - \{T\}$ there exists a word $w \in S_T$ such that $T'$ assigns probability zero to $w$. It can be checked that $p$-separability is $(p, 1)$-separability and that the matrix scenario is $(p, 2)$-separable (with $p = n^{-2}$ for $n \times n$ matrices, $n \geq 2$). In practice, $(p, t)$-separability captures the notion that every topic is uniquely identified by a set of $t$ words. We shall refer to these sets as *anchor sets*. With this terminology, $p$-separability is just $(p, 1)$-separability with singleton anchor sets.

In this paper we give an algorithm for LDA topic reconstruction (under $(p, 1)$-separability) that, starting from a random LDA corpus over a vocabulary of $m$ words consisting of

$$\Theta\left(\frac{K^2 \cdot \max\left(1, K^2 \alpha^2\right)}{\delta^2 \cdot p^2} \cdot \log m\right)$$

many documents of (at least) 2 words each, computes a set of topics $\mathcal{T}'$ which is an approximation of the true set of topics $\mathcal{T}$ with error $\delta$ (in $\ell_\infty$-norm)[4]. Asymptotically, this compares favourably to the bound of Equation (1) but it is also the case that the algorithm is very simple and efficient. The Dirichlet parameter $\alpha$ is typically assumed to be $O(1/K)$, in which case the term $\max\left(1, K^2\alpha^2\right)$ resolves to a constant, and the number of documents required for reconstruction becomes $\Theta\left(\frac{K^2}{\delta^2 \cdot p^2} \cdot \log m\right)$.

Note that the Dirichlet distribution is such that, as $\alpha$ goes to zero, the admixture becomes more and more polarized, in the sense that the documents resemble more and more single-topic documents, which intuitively facilitates topic reconstruction. When $\alpha$ moves in the other direction toward larger and larger values, the admixture creates documents in which all topics are equally represented, which makes reconstruction more expensive in the sense that the size of corpora must become bigger and bigger. These considerations apply to all algorithms, but we note that our dependence on $K$ and $p$ is much milder than those of the other algorithms we are discussing.

It is interesting to compare the overall structure of our algorithm to that in Arora *et al.* (2012). The first step of the latter is to project points into a low-dimensional space, where computation is cheaper, by preserving distances. In a very loose sense, this is equivalent to our reduction, which transforms the distribution of documents of length 2 from LDA to STA. The second step is a very natural one: try to identify the anchor words, using a simple combinatorial procedure (or, more generally, the $t$-anchor sets, starting from documents of length $t + 1$). The third step is again very natural: use the anchors to build the topics. It is here that the full advantage of our approach becomes evident. Our algorithm attempts the reconstruction in the single topic world — a much less daunting prospect than reconstruction in the full-fledged LDA world. As a result, our third step is a very simple procedure — in the LDA world one would have had to pay the price of heavy-duty linear algebra computations.

In order to deal with $(p, t)$-separable topics the algorithm only needs documents of length $t + 1$. Therefore, in order to reconstruct $p$-separable topics ($t = 1$) it only needs bigrams, and in the matrix case ($t = 2$) only trigrams! Clearly, this has a significant positive impact on efficiency.

We also present a comparative experimental evaluations which shows that our approach compares favorably to those of (Arora *et al.* , 2012, 2013; Griffiths & Steyvers, 2004; Anandkumar *et al.* , 2014).

The paper is organized as follows. We start in § 2 with some quick preliminaries. In § 3 we give the reduction from LDA to STA, followed by § 4 in which a robust algorithm for STA topic reconstruction is presented with which we solve the $(p, t)$-separable case for $t = 1, 2$, which subsumes both $p$-separability and matrix-like topics. § 5 presents our experiments. The proofs missing from the main body of the paper can be found in the Supplementary Material archive.

## 2  Preliminaries

Throughout the paper, we will use $\mathcal{V}$ to denote the underlying vocabulary and assume without loss of generality that $m := |\mathcal{V}| \geq 2$, since the case $m = 1$ is trivial (there can be only one topic).

We will only deal with LDA when the admixtures come from a symmetric Dirichlet distribution whose parameter will be denoted by $\alpha$. Since this is the only case we consider and there is no danger of confusion, we will sometimes omit to specify that we are dealing with symmetric LDA.

We will use the following notation. Given a set of $K$ topics $\mathcal{T}$ and a Dirichlet parameter $\alpha$, $\mathcal{D}_\ell^{\mathcal{T}}$ will denote the distribution induced by LDA over the topics of length $\ell$. When there is no danger for confusion subscripts and superscripts will be dropped. Similarly, $\mathcal{S}_\ell^{\mathcal{T}}$ will refer to the distribution induced by STA over the topics of length $\ell$. And, likewise, subscripts and superscripts will be dropped when no danger for confusion may arise.

## 3  A Reduction from LDA to STA

In this section we give the approximation preserving reduction from LDA to STA. As usual, in the background we have a set of unknown topics $\mathcal{T}$ that we wish to reconstruct. The reduction takes as input the Dirichlet parameter $\alpha$, an approximation $\widetilde{\mathcal{D}}$ of the document distribution $\mathcal{D}$ generated by LDA with topics $\mathcal{T}$, and gives as output an approximation $\widetilde{\mathcal{S}}$ of the document distribution $\mathcal{S}$ generated by STA with the same set of topics $\mathcal{T}$. The point of departure is a reduction between the two true distributions $\mathcal{D}$ and $\mathcal{S}$ established by (Chierichetti *et al.* , 2018, Section 4).

**Definition 1.** *Given a permutation $\pi \in \text{Sym}([\ell])$, let $\mathcal{C}_\pi$ be the partition of $[\ell]$ into the cycles of $\pi$:*

$$\mathcal{C}_\pi = \left\{ S \mid S \subseteq [\ell] \text{ and the elements of } S \text{ form a cycle in } \pi \right\}.$$

*Furthermore, for $d \in \mathcal{V}^\ell$ and $S = \{i_1, i_2, \ldots, i_{|S|}\} \subseteq [\ell]$ with $i_1 < i_2 < \ldots < i_{|S|}$, let $d_{|S}$ be the document containing the words $d(i_1), \ldots, d(i_{|S|})$ in this order (that is, let it be the document that is obtained by removing from $d$ the words whose positions in $d$ are not in $S$).*

For example, if $\pi = (163)(25)(4)$ then $\mathcal{C}_\pi = \{\{1, 3, 6\}, \{2, 5\}, \{4\}\}$. And, if $d = w_1 w_2 w_3 w_4 w_5 w_6$ and $S = \{1, 3, 6\}$ then $d_{|S} = w_1 w_3 w_6$.

**Theorem 2** (Reduction from LDA to STA (Chierichetti *et al.* , 2018))**.** *Let $\mathcal{T}$ be any set of $K$ topics on a vocabulary $\mathcal{V}$ and consider any $d \in \mathcal{V}^\ell$. Then, for any $\alpha > 0$,*

$$\mathcal{S}_\ell^{\mathcal{T}}(d) = \frac{\Gamma(K \cdot \alpha + \ell)}{\Gamma(K \cdot \alpha + 1) \cdot \Gamma(\ell)} \cdot \mathcal{D}_\ell^{\mathcal{T},\alpha}(d) - \frac{1}{K \cdot \alpha \cdot \Gamma(\ell)} \cdot \sum_{\substack{\pi \in \mathrm{Sym}([\ell]) \\ |C_\pi| \geq 2}} \prod_{S \in C_\pi} \left( K \cdot \alpha \cdot \mathcal{S}_{|S|}^{\mathcal{T}}(d_{|S}) \right). \quad (2)$$

Equation (2) looks rather formidable, but the point is that it can be taken as a blackbox to transform one probability distribution into the other. Note that the equation is recursive — it specifies how to compute the STA distribution $\mathcal{S}_\ell$ over documents of length $\ell$, from $\mathcal{D}_\ell$ and the STA distributions $\mathcal{S}_1, \ldots, \mathcal{S}_{\ell-1}$ over documents of length less than $\ell$. In the base case — documents of length one — the two distributions $\mathcal{D}_1$ and $\mathcal{S}_1$ coincide and thus the induction can be kick-started.

The next lemma tells us how to compute a good approximation $\widetilde{\mathcal{D}}$ of the true document distribution $\mathcal{D}$ induced by LDA starting from a corpus.

**Lemma 3** (LDA Probabilities Approximation)**.** *Fix $\ell \geq 1$, and $\xi \in (0, 1)$. Let $X_1, \ldots, X_n$ be $n$ iid samples from $\mathcal{D}_\ell^{\mathcal{T},\alpha}$. For $i \in [\ell]$, and for a document $d \in [m]^i$, let $n_d$ be the number of samples having $d$ as a prefix, $n_d = |\{j | j \in [n] \land d \text{ is a prefix of } X_j\}|$. For $i \in [\ell]$, and for a document $d \in [m]^i$, let $\widetilde{\mathcal{D}}_i(d) = \frac{n_d}{n}$ be the empirical fraction of the samples whose $i$-prefix is equal to $d$. Then,*

(a) *If $n \geq \left\lceil \frac{2}{\xi^2} \cdot \ell \cdot \ln m \right\rceil$, with probability at least $1 - O(m^{-\ell})$, for every document $d$ of length $i \leq \ell$, it holds that $|\mathcal{D}_i^{\mathcal{T},\alpha}(d) - \widetilde{\mathcal{D}}_i(d)| \leq \xi$.*

(b) *For any $q > 0$, if $n \geq \left\lceil \frac{9}{q \cdot \xi^2} \cdot \ell \cdot \ln m \right\rceil$, with probability at least $1 - O(m^{-\ell})$, for every document $d$ of length $i \leq \ell$ such that $\mathcal{D}_i^{\mathcal{T},\alpha}(d) \geq q$, it holds that $\widetilde{\mathcal{D}}_i(d) = (1 \pm \xi)\mathcal{D}_i^{\mathcal{T},\alpha}(d)$.*

The next theorem establishes our main result of this section, namely that Equation (2) is approximation preserving.

**Theorem 4** (Single-Topic Probabilities Approximation)**.** *Fix $\xi \in (0, 1)$. Given an approximation $\widetilde{\mathcal{D}}_i(d)$ of $\mathcal{D}_i^{\mathcal{T},\alpha}(d)$, $i \in \{1, 2\}$, define $\widetilde{\mathcal{S}}_1 = \widetilde{\mathcal{D}}_1$, and $\widetilde{\mathcal{S}}_2(ww') = (K\alpha + 1) \cdot \widetilde{\mathcal{D}}_2(ww') - K\alpha \cdot \widetilde{\mathcal{S}}_1(w) \cdot \widetilde{\mathcal{S}}_1(w')$. Then,*

(a) *If for every document $d$ of length $i \leq 2$ it holds $|\mathcal{D}_i^{\mathcal{T},\alpha}(d) - \widetilde{\mathcal{D}}_i(d)| \leq \frac{\xi}{4(K\alpha+1)}$, then $|\mathcal{S}_i^{\mathcal{T}}(d) - \widetilde{\mathcal{S}}_i(d)| \leq \xi$.*

(b) *If, for a given word $w$, it holds $\widetilde{\mathcal{D}}_1(w) = \left(1 \pm \frac{\xi}{4K\alpha+1}\right) \mathcal{D}_1^{\mathcal{T},\alpha}(w)$ and $\widetilde{\mathcal{D}}_2(ww) = \left(1 \pm \frac{\xi}{4K\alpha+1}\right) \mathcal{D}_2^{\mathcal{T},\alpha}(ww)$, then $\widetilde{\mathcal{S}}_2(ww) = (1 \pm \xi)\mathcal{S}_2^{\mathcal{T}}(ww)$.*

## 4 Robust Algorithms for STA Topic Identifiability

In this section we give an algorithm for identifying $p$-separable topics (or, equivalently, $(p, 1)$-separable topics). As usual, we have a set $\mathcal{T}$ of topics in the background that we wish to identify.

The first step is to identify anchor words or their proxies. By proxy, or quasi-anchor word, we mean that the word has "large" probability in one topic and very small probabilities in the remaining ones.

We begin with a technical lemma stating that if a vector has a coordinate that is very large with respect to the others, then all of its $\ell_p$-norms are close to one another. Loosely speaking, the lemma says that if a word is an anchor word or a quasi-anchor word then, if we look at the vector consisting of the probabilities assigned to this word by the topics, the $\ell_p$-norms of the vector are close.

**Lemma 5.** *Let $v \in \mathbf{R}^n$, and suppose that $|v|_\infty = (1 - \epsilon) \cdot |v|_1$, for some $\epsilon \in [0, 1)$. Then, for each $p \geq 1$, $(1 - \epsilon)^p \cdot |v|_1^p \leq |v|_p^p \leq (1 - \epsilon)^{p-1} \cdot |v|_1^p$.*

The next theorem tells us how to spot anchor words. The idea is that if a word $w$ is an anchor word then there is a signal telling us so. Consider the two documents $w$ and $ww$. The signal is the ratio $\mathcal{S}_2^{\mathcal{T}}(ww)/K\, \mathcal{S}_1^{\mathcal{T}}(w)^2$. If $w$ is an anchor word this ratio equals 1, and if $w$ is "far" from being an anchor word then the ratio is bounded below 1. In fact, the theorem tells us more. If we have two good approximations $\widetilde{\mathcal{S}}_1(w)$ and $\widetilde{\mathcal{S}}_2(ww)$ of, respectively, $\mathcal{S}_1^{\mathcal{T}}(w)$ and $\mathcal{S}_2^{\mathcal{T}}(ww)$ then the ratio $\rho_w = \widetilde{\mathcal{S}}_2(ww)/K\, \widetilde{\mathcal{S}}_1(w)^2$ will have (approximately) the same properties. Since we are dealing with an approximation of the true distribution $\mathcal{S}$, this tells us that we will be able to spot anchors even in this case.

Now, fix a word $w$ of the dictionary let $x_w$ be the (unknown) vector of its probabilities in the $K$ topics, so that $\mathcal{S}_1^{\mathcal{T}}(w) = K^{-1} \cdot |x_w|_1$ and $\mathcal{S}_2^{\mathcal{T}}(ww) = K^{-1} \cdot |x_w|_2^2$.

**Theorem 6.** *Let $\xi \in (0, 1)$ and $w \in \mathcal{V}$ be any word. Suppose that $\widetilde{\mathcal{S}}_1(w) = (1 \pm \xi)\mathcal{S}_1^{\mathcal{T}}(w)$ and $\widetilde{\mathcal{S}}_2(ww) = (1 \pm \xi)\mathcal{S}_2^{\mathcal{T}}(ww)$. Define $\rho_w = \frac{\widetilde{\mathcal{S}}_2(ww)}{K\,(\widetilde{\mathcal{S}}_1(w))^2}$.*

*Then, if $\epsilon_w$ is such that $|x_w|_\infty = (1 - \epsilon_w) \cdot |x_w|_1$, it holds $\frac{(1-\epsilon_w)^2(1-\xi)}{(1+\xi)^2} \leq \rho_w \leq \frac{(1-\epsilon_w)(1+\xi)}{(1-\xi)^2}$.*

Consider the quantity $\rho_w$ defined by the previous theorem and suppose that $\rho_w \geq \frac{1-\xi}{(1+\xi)^2}$. The next lemma says that if $w$ is an anchor word, then $\rho_w$ satisfies the inequality. And, viceversa, if $\rho_w$ satisfies it, then $w$ must be either an anchor word or a quasi-anchor word, which can also be used for topic reconstruction.

**Lemma 7.** *Let $\xi \in (0, 1)$. Suppose that $\widetilde{\mathcal{S}}_1(w) = (1 \pm \xi)\mathcal{S}_1^{\mathcal{T}}(w)$ and $\widetilde{\mathcal{S}}_2(ww) = (1 \pm \xi)\mathcal{S}_2^{\mathcal{T}}(ww)$. Let $\rho_w = \frac{\widetilde{\mathcal{S}}_2(ww)}{K\,(\widetilde{\mathcal{S}}_1(w))^2}$, and $\epsilon_w$ be such that $|x_w|_\infty = (1 - \epsilon_w) \cdot |x_w|_1$.*

*If $\epsilon_w = 0$ then $\rho_w \geq \frac{1-\xi}{(1+\xi)^2}$. Moreover, if $\rho_w \geq \frac{1-\xi}{(1+\xi)^2}$ then $\epsilon_w \leq 6\xi$.*

The previous lemma gives us a simple test to identify anchor words or quasi-anchor words. We know that each anchor word is uniquely associated with one topic — the one that assigns to it non zero probability. We will see later that $\xi$ can be chosen in a way that quasi-anchor words too can be associated with one topic — the one assigning it a much larger probability than the other topics. The next lemma tells us how to determine whether two different words insist on the same topic.

We say that a topic $j$ is *dominant* for a word $w$, if (i) $w$ has a unique largest probability in the topics, and (ii) its largest probability is in topic $j$. We say that the words $w, w'$ are codominated, if there exists a topic $j$ such that $j$ is dominant for both $w$ and $w'$.

**Theorem 8.** *For $w \in \{w_1, w_2\}$, suppose that $\widetilde{\mathcal{D}}_1(w) = (1 \pm \xi)\mathcal{D}_1^{\mathcal{T}}(w)$, and that $|x_w|_\infty = (1 - \epsilon_w) \cdot |x_w|_1$. Suppose further that $\widetilde{\mathcal{D}}_2(w_1 w_2) = (1 \pm \xi)\mathcal{D}_2^{\mathcal{T}}(w_1 w_2)$.*

*Define $\tau(w_1, w_2) := \frac{\widetilde{\mathcal{D}}_2(w_1 w_2)}{\widetilde{\mathcal{D}}_1(w_1) \cdot \widetilde{\mathcal{D}}_1(w_2)}$. If the words $w_1$ and $w_2$ are co-dominated, then*

$$\tau(w_1, w_2) \geq \frac{(1 - \xi)}{(1 + \xi)^2} \cdot \frac{K\alpha + K(1 - \epsilon_1)(1 - \epsilon_2)}{K\alpha + 1},$$

*otherwise*

$$\tau(w_1, w_2) \leq \frac{(1 + \xi)}{(1 - \xi)^2} \frac{K\alpha + K(\epsilon_{w_1} + \epsilon_{w_2} + \epsilon_{w_1}\epsilon_{w_2})}{K\alpha + 1}.$$

The next corollary gives a simple way to determine which quasi-anchor words belong to the same topic.

**Corollary 9.** *Let $A$, $|A| > K$, be a set of quasi-anchor words $w$ with $|x_w|_\infty = (1 - \epsilon_w) \cdot |x_w|_1$. Let $\xi < \frac{1}{6}\frac{1-4\epsilon}{\alpha+1}$, where $\epsilon = \max_{w \in A} \epsilon_w$. Suppose that $\widetilde{\mathcal{D}}_1(w) = (1 \pm \xi)\mathcal{D}_1^{\mathcal{T}}(w)$ for $w \in A$, and let $E$ be the maximal subset of $\binom{A}{2}$ such that $\widetilde{\mathcal{D}}_2(w_1 w_2) = (1 \pm \xi)\mathcal{D}_2^{\mathcal{T}}(w_1 w_2)$ for each $\{w_1, w_2\} \in E$.*

*If $E$ contains all the co-dominated pairs of words, the correct partitioning of $A$ according to the $K$ topics $\mathcal{T}$ can be obtained by iteratively assigning to the same group the pair of words $\{w_1, w_2\} \in E$ with largest $\tau(w_1, w_2) := \frac{\widetilde{\mathcal{D}}_2(w_1 w_2)}{\widetilde{\mathcal{D}}_1(w_1) \cdot \widetilde{\mathcal{D}}_1(w_2)}$ until reaching $K$ groups.*

We then have the main theorem of this section which gives the full algorithm for topic reconstruction in the $p$-separable (equivalent to the $(p, 1)$-separable) case.

**Theorem 10** (Main Result). *Suppose that $\mathcal{T}$ is a set of $K = |\mathcal{T}|$ topics, and let $\delta \leq 1/48$. There exists an algorithm that, under the $p$-separability assumption, and under the LDA model $\mathcal{D}^{\mathcal{T},\alpha}$, with probability $1 - o(1)$ reconstructs each topic in $\mathcal{T}$ to within an $\ell_\infty$ additive error upper bounded by $\delta$, by accessing $n = \Theta\left( \frac{K^2 \cdot \max\left((K\alpha)^2, 1\right)}{\delta^2 p^2} \cdot \ln m \right)$ iid samples from $\mathcal{D}_2^{\mathcal{T},\alpha}$. The algorithm runs in $O(n)$.*

Algorithm 1 is a version of the method analyzed in Theorem 10. The most notable feature of our algorithm is its simplicity.

---

**Algorithm 1** The Algorithm for reconstructing $(p, 1)$-separable topics.

---

**Require:** $K, p > 0, \delta$, corpus $\mathcal{C}$ of documents, $\alpha$ parameter of the symmetric LDA mixture,
1: Let $W$ be the set of words $w$ whose empirical fraction in $\mathcal{C}$ is at least $p/2K$.
2: For each $w, w' \in W$, estimate the empirical fraction of the document $ww'$ in $\mathcal{C}$ — that is, obtain approximations $\widetilde{\mathcal{D}}_1$ and $\widetilde{\mathcal{D}}_2$ of $\mathcal{D}_1$ and $\mathcal{D}_2$ .
3: Apply the reduction of Theorem 2 to estimate the uniform single-topic probabilities $\widetilde{\mathcal{S}}_1(w)$ and $\widetilde{\mathcal{S}}_2(ww')$.
4: For each $w \in W$, compute $\rho_w := \frac{\widetilde{\mathcal{S}}_2(ww)}{K\,(\widetilde{\mathcal{S}}_1(w))^2}$ and add $w$ to the set $A$ of *quasi-anchors* if $\rho_w \geq \frac{1-\delta}{(1+\delta)^2}$.
5: Use Corollary 9 on $A$ to obtain $K$ pairwise non-codominated quasi-anchor words $w_1, w_2, \ldots, w_K$.
6: For each $w_i$, return a topic whose probability on word $w \in \mathcal{V}$ is $\widetilde{\mathcal{S}}_2(w_i w)/\widetilde{\mathcal{S}}_1(w_i)$.

---

### 4.1 The general $(p, t)$-separable case

The algorithm we have developed in the previous section can be generalized to work for $(p, t)$-separable topics (this is what we need to deal with the topic structure of Griffiths & Steyvers (2004)). The generalization is quite straightforward and is a natural extension of Algorithm 1 but, for lack of space, we defer it to the full paper. We will however compare our generalized algorithm to Gibbs sampling — the method used by Griffiths & Steyvers (2004) — in the next section.

## 5 Experimental Results

We compare our approach[5] to three state-of-the-art algorithms: GIBBS sampling[6], a popular heuristic approach, the algorithm from (Arora *et al.* , 2013) for $p$-separable instances, referred to as RECOVER from now on, and the implementation of Yau (2018) of the tensor-based algorithm (henceforth TENSOR) introduced in (Anandkumar *et al.* , 2014). Each of these algorithms was executed on the same computer: an Intel Xeon CPU E5-2650 v4, 2.20GHz, with 64GiB of DDR4 RAM. We used a single core per algorithm.

**The topics.** For the experiments we generated a family of $k$ topics in various ways, for $k = 10, 25, 50$. The family NIPS TOPICS was generated by running Gibbs sampling on the NIPS dataset (Newman, 2008). Since these topics are not $p$-separable in general, a second family was generated by adding anchor words artificially. A third family, SYNTHETIC, was generated by sampling from a uniform Dirichlet distribution with parameter $\beta = 1$ and, to enforce $p$-separability, anchor words were added. Finally, a fourth family of topics were GRID topics. These are the prototypical grid-like topics of sizes $7 \times 7$ and $5 \times 5$ (introduced by Griffiths & Steyvers (2004)); notice that these are $(p, 2)$-separable but not $(p, 1)$-separable.

In each instance except grid topics, the number of words of the vocabulary was set to $m = 400$.

**The corpora.** From each one of the set of topics specified above, we generated a corpus of $n$ documents of length $\ell$, for $n = 10^4, 10^5, 10^6$ and $\ell = 2, 3, 10, 100$. Because of space constraints, we will only show results for $n = 10^6$.

We are interested in two aspects of performance, the wall-clock running time and quality of the reconstruction, measured as the $\ell_\infty$ norm between the true set of topics and the reconstruction. To assess this, we computed the best possible matching between the two families of topics as follows. Consider a bipartite graph with the true set of topics on one side of the bipartition and the reconstructed topics on the other. Between every pair of topics on opposite sides, there is an edge of weight equal to their $\ell_\infty$ distance. The quality of the reconstruction is given by the minimum cost perfect matching in this graph. All algorithms were run on a single thread.

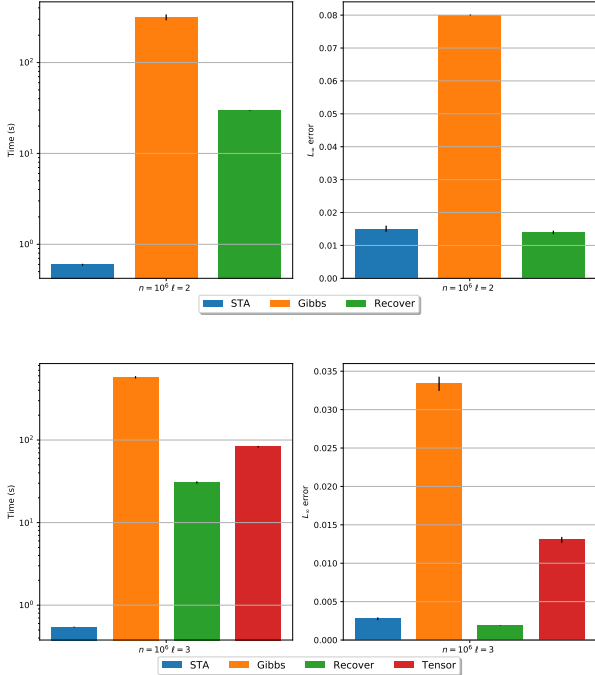

Figure 1: The top-left plot shows the wall clock time (in seconds, on a log-scale) required by the algorithms with NIPS TOPICS, with documents of length $\ell = 2$ and 10 topics (TENSOR is not shown since it requires $\ell > 2$). The top-right plot shows (on a linear-scale) the $\ell_\infty$ error of the algorithms on the same instance; observe that STA is faster than the other two algorithms by more than one order of magnitude, while its error is almost as good as that of RECOVER. The bottom-left plot shows the wall clock time (in seconds, on a log-scale) required by the algorithms with 10 SYNTHETIC topics, with documents of length $\ell = 3$. As before, STA is faster than the other algorithms by more than one order of magnitude and its error is almost as good as the one of RECOVER.

Conceptually our algorithm implements the following pipeline, $\mathcal{C} \xrightarrow{(1)} \mathcal{L} \xrightarrow{(2)} \mathcal{S} \xrightarrow{(3)} \mathcal{T}$, where the first step, starting from the corpus $\mathcal{C}$, computes the approximation to the distribution induced over the documents by LDA; the second step implements the reduction from the latter to the STA-induced distribution, and, lastly, the third step is Algorithm 1. We implemented the steps of this pipeline with several optimizations. In particular, we did not fully compute the approximate distributions $\mathcal{L}$ and $\mathcal{S}$: rather, we lazily computed their entries that were requested by Algorithm 1.

Algorithm 1 simply picks the first two words of a document and throws the rest away, seemingly a rather wasteful thing to do. A natural alternative is to feed the algorithm we all pairs of words from the document, hoping that the correlations so introduced can be safely ignored. This variant, which we call STA in the following, was consistently more accurate than Algorithm 1 at the expense of a small increase in the running time. Therefore this is the implementation that we discuss.

In the case of grid-like topics, STA is the version of Algorithm 1 for the $(p, 2)$-separable case.

**Wall-clock time.** The two plots on the left of Figure 1 compare the running times of the algorithms with corpora of documents of length $\ell = 2, 3$, with 10 topics. As expected, STA for documents of

length 2 is much faster then the other algorithms (while GIBBS is especially cumbersome), and its reconstruction quality is close to the best one. This figure exemplifies the general picture: a similar outcome was observed for all values of $n$ topic families.

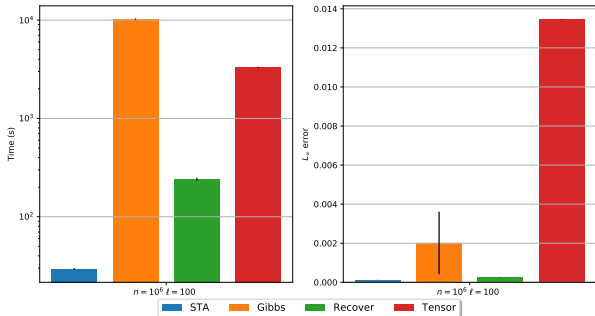

Figure 2: On the left: wall clock time (in seconds, on a log-scale) required by the algorithms with 10 SYNTHETIC topics, with documents of length $\ell = 100$. On the right: the $\ell_\infty$ error (on a linear-scale) of the algorithms on the same instance.

**Precision of the reconstruction.** Figure 1 exemplifies the general picture that emerges from our tests, for short documents and 1-separable topics. RECOVER and STA have the smallest reconstruction errors. As expected, GIBBS did not work well with very short documents. Therefore we tested the algorithms with documents of length $\ell = 100$. In Figure 2, we show that STA gives the best reconstruction, and its the fastest one by at least one order of magnitude.

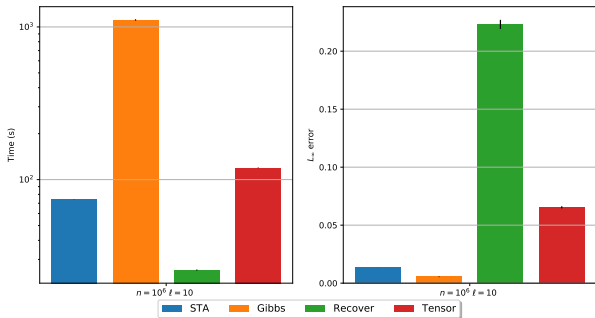

Figure 3: On the left: wall clock time (in seconds, on a log-scale) required by the algorithms on a $5 \times 5$ GRID with 10 topics, with documents of length $\ell = 10$. On the right: the $\ell_\infty$ error (on a linear-scale) of the algorithms on the same instance. Recall that, here, STA is the version of Algorithm 1 for $(p, 2)$-separability. On this instance, RECOVER is the fastest algorithm; observe, though, that RECOVER returns topics that are *very* far from the original ones, since this instance is not $p$-separable.

**Grid.** In a final set of experiments, we considered the prototypical GRID instances of sizes $7 \times 7$ and $5 \times 5$ (introduced in Griffiths & Steyvers (2004)). In Figure 3, we see that STA and GIBBS provide an $\ell_\infty$ error smaller by an order of magnitude than that of RECOVER (and 4 times smaller than that of Tensor). Moreover, the running time of STA is at least one order of magnitude smaller than that of GIBBS.

**Assessment.** A picture emerges from our experiments. STA offers a pretty good reconstruction, while being extremely competitive in terms of running time. We see this as an encouraging proof of concept that warrants further investigation of the approach introduced in this paper, that is, reducing LDA-reconstruction to the much simpler problem of STA-reconstruction. A more careful implementation of our algorithms could further increase the speed of our approach, while more ideas seem to be needed to improve the quality of reconstruction. Our experiments show that this could be a worthwhile endeavor.

## Footnotes

[4]More precisely, there exists a bijection $\phi : \mathcal{T} \to \mathcal{T}'$ such that, for each $T \in \mathcal{T}$, $|T - \phi(T)|_\infty \leq \delta$

[5]Our implementation can be downloaded from `https://github.com/matteojug/lda-sta`.

[6]We use the popular MALLET library McCallum (2002), `http://mallet.cs.umass.edu/`, with a 200 iteration burnin period and 1000 iterations.

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
