[Supplementary Material]

# A Reduction for Efficient LDA Topic Reconstruction
# Supplementary Material

**Matteo Almanza**[*]
Sapienza University
Rome, Italy
almanza@di.uniroma1.it

**Flavio Chierichetti**[†]
Sapienza University
Rome, Italy
flavio@di.uniroma1.it

**Alessandro Panconesi**[‡]
Sapienza University
Rome, Italy
ale@di.uniroma1.it

**Andrea Vattani**
Spiketrap
San Francisco, CA, USA
avattani@cs.ucsd.edu

## A   Missing Proofs

### A.1   Proof of Lemma 3

Observe that, by the bag of words property, the prefix of length $i$ of a sample from $\mathcal{D}_\ell^{\mathcal{T},\alpha}$, $\ell \geq i$, is distributed like a sample from $\mathcal{D}_i^{\mathcal{T},\alpha}$.

By the Hoeffding bound, we know that if $Z = \sum_{i=1}^n Z_i$ is that sum of $n$ iid $Z_1, \ldots, Z_n$ satisfying $0 \leq Z_i \leq 1$, then (a) $\Pr[|Z - E[Z]| \geq n\xi] \leq 2e^{-2n\xi^2}$; and (b) $\Pr[|Z - E[Z]| \geq \xi E[Z]] \leq 2e^{-E[Z]\xi^2/3}$. Take a given document $d \in [m]^i$ of length $i \in [\ell]$ and observe that $\mathcal{D}_i^{\mathcal{T},\alpha}(d) = E[n_d/n]$. By applying (a), if $n \geq \frac{2}{\xi^2} \cdot \ell \cdot \ln m$,

$$\Pr[|\mathcal{D}_i^{\mathcal{T},\alpha}(d) - \widetilde{\mathcal{D}}_i(d)| \geq \xi] = \Pr\left[\left|n \cdot \mathcal{D}_i^{\mathcal{T},\alpha}(d) - n_d\right| \geq n\xi\right] \leq 2e^{-2n\xi^2} \leq 2e^{-4\ell \ln m} = 2m^{-4\ell}.$$

Similarly, using (b), if $\mathcal{D}_i^{\mathcal{T},\alpha}(d) \geq q$ and $n \geq \frac{9}{q \cdot \xi^2} \cdot \ell \cdot \ln m$,

$$\Pr[|\mathcal{D}_i^{\mathcal{T},\alpha}(d) - \widetilde{\mathcal{D}}_i(d)| \geq \xi\mathcal{D}_i^{\mathcal{T},\alpha}(d)] \leq 2e^{-nq\xi^2/3} \leq 2e^{-3\ell \ln m} = 2m^{-3\ell}.$$

The number of documents of length at most $[\ell]$ is upper bounded by

$$\sum_{i=0}^{\ell} m^i \leq \frac{m^{\ell+1} - 1}{m - 1} \leq m^{\ell+1}.$$

By union bounding across all the documents of length smaller than or equal $\ell$, we get the stated claim.

### A.2   Proof of Theorem 4

First we prove the following technical Lemma, that will later be used in the proof of Theorem 4.

[*]Supported in part by the ERC Starting Grant DMAP 680153, and by the "Dipartimenti di Eccellenza 2018-2022" grant awarded to the Dipartimento di Informatica at Sapienza.

[†]Supported in part by the ERC Starting Grant DMAP 680153, by a Google Focused Research Award, and by the "Dipartimenti di Eccellenza 2018-2022" grant awarded to the Dipartimento di Informatica at Sapienza.

[‡]Supported in part by the ERC Starting Grant DMAP 680153, by a Google Focused Research Award, and by the "Dipartimenti di Eccellenza 2018-2022" grant awarded to the Dipartimento di Informatica at Sapienza.

**Lemma 1.** *Let* $\lambda = \lambda(\alpha, K, \mathcal{T}) = \frac{\left(\mathcal{S}_1^{\mathcal{T}}(w)\right)^2}{\mathcal{D}_2^{\mathcal{T},\alpha}(ww)}$. *Then it holds that* $0 \leq \lambda \leq 1$.

*Proof.* We now show that $0 \leq \lambda \leq 1$. The lower bound is trivial; we prove the upper bound. Fix a word $w$, and let $x_w$ be the vector of its probabilities in the $K$ topics, so that $\mathcal{S}_1^{\mathcal{T}}(w) = K^{-1} \cdot |x_w|_1$ and $\mathcal{S}_2^{\mathcal{T}}(ww) = K^{-1} \cdot |x_w|_2^2$. By applying Theorem 2, we can rewrite $\mathcal{D}_2^{\mathcal{T},\alpha}(ww)$ as

$$\mathcal{D}_2^{\mathcal{T},\alpha}(ww) = \frac{1}{K\alpha+1}\mathcal{S}_2^{\mathcal{T}}(ww) + \frac{K\alpha}{K\alpha+1}\mathcal{S}_1^{\mathcal{T}}(w)^2 = \frac{1}{K^2\alpha+K}|x_w|_2^2 + \frac{\alpha}{K^2\alpha+K}|x_w|_1^2.$$

Then,

$$\frac{\left(\mathcal{S}_1^{\mathcal{T}}(w)\right)^2}{\mathcal{D}_2^{\mathcal{T},\alpha}(w,w)} = \frac{K^{-2}\cdot|x_w|_1^2}{\frac{1}{K^2\alpha+K}(|x_w|_2^2 + \alpha|x_w|_1^2)} = \left(\alpha+\frac{1}{K}\right)\cdot\frac{|x_w|_1^2}{|x_w|_2^2 + \alpha|x_w|_1^2} = \left(\alpha+\frac{1}{K}\right)\cdot\frac{1}{\frac{|x_w|_2^2}{|x_w|_1^2}+\alpha}.$$

The vector $x_w$ has $K$ dimension. Thus, by the Cauchy-Schwartz inequality, we have that $|x_w|_1^2 \leq K \cdot |x_w|_2^2$, and

$$\frac{\left(\mathcal{S}_1^{\mathcal{T}}(w)\right)^2}{\mathcal{D}_2^{\mathcal{T},\alpha}(w,w)} \leq \left(\alpha+\frac{1}{K}\right)\cdot\frac{1}{\frac{|x_w|_2^2}{K\cdot|x_w|_2^2}+\alpha} = 1. \quad \square$$

We now move on to the proof of Theorem 4. By Theorem 2, we have that $\mathcal{S}_1^{\mathcal{T}}(w) = \mathcal{D}_1^{\mathcal{T}}(w)$ for each $w \in [m]$ and $\mathcal{S}_2^{\mathcal{T}}(ww') = (K\alpha+1)\cdot\mathcal{D}_2^{\mathcal{T}}(ww') - K\alpha\cdot\mathcal{S}_1^{\mathcal{T}}(w)\cdot\mathcal{S}_1^{\mathcal{T}}(w')$. Let $D_i = \max_{d \in [m]^i} |\mathcal{D}_i^{\mathcal{T},\alpha}(d) - \widetilde{\mathcal{D}}_i(d)|$ and $S_i = \max_{d \in [m]^i} |\mathcal{S}_i^{\mathcal{T}}(d) - \widetilde{\mathcal{S}}_i(d)|$. Observe that, since $\widetilde{\mathcal{S}}_1 = \widetilde{\mathcal{D}}_1$ and $\mathcal{S}_1^{\mathcal{T}} = \mathcal{D}_1^{\mathcal{T},\alpha}$, it holds that $S_1 = D_1 \leq \frac{\xi}{4K\alpha+4}$.

We now proceed to bound $\mathcal{S}_2$, in terms of $\mathcal{D}_2$ and $\mathcal{S}_1$. First, we observe that, in general, if it holds $0 \leq x_j \leq 1$, and $0 \leq \epsilon_j \leq 1$, for each $j \in [n]$, then

$$|(x_1+\epsilon_1)\cdot(x_2+\epsilon_2) - x_1 x_2| \leq 3\cdot\max(|\epsilon_1|, |\epsilon_2|).$$

Thus,

$$\left|\tilde{\mathcal{S}}_1(w)\cdot\tilde{\mathcal{S}}_1(w') - \mathcal{S}_1^{\mathcal{T}}(w)\cdot\mathcal{S}_1^{\mathcal{T}}(w')\right| \leq 3\cdot S_1 < \frac{3\xi}{4K\alpha}.$$

We now compute $S_2$:

$$\left|\mathcal{S}_2^{\mathcal{T}}(ww') - \widetilde{\mathcal{S}}_2(ww')\right| = \left|(K\alpha+1)\left(\mathcal{D}_2^{\mathcal{T}}(ww') - \widetilde{\mathcal{D}}_2(ww')\right) - K\alpha\left(\mathcal{S}_1^{\mathcal{T}}(w)\mathcal{S}_1^{\mathcal{T}}(w') - \widetilde{\mathcal{S}}_1(w)\widetilde{\mathcal{S}}_1(w')\right)\right|$$

$$\leq (K\alpha+1)\left|\mathcal{D}_2^{\mathcal{T}}(ww') - \widetilde{\mathcal{D}}_2(ww')\right| + K\alpha\left|\mathcal{S}_1^{\mathcal{T}}(w)\mathcal{S}_1^{\mathcal{T}}(w') - \widetilde{\mathcal{S}}_1(w)\widetilde{\mathcal{S}}_1(w')\right|$$

$$< (K\alpha+1)\cdot\frac{\xi}{4\cdot(K\alpha+1)} + K\alpha\cdot\frac{3\xi}{4K\alpha} = \xi,$$

and the proof of the first claim is complete.

We now proceed to the second claim. Let $\lambda := \lambda(w, \alpha, K, \mathcal{T}) = \frac{\left(\mathcal{S}_1^{\mathcal{T}}(w)\right)^2}{\mathcal{D}_2^{\mathcal{T},\alpha}(ww)}$. By Lemma 1, we know that $0 \leq \lambda \leq 1$. By Theorem 2, we have that

$$\mathcal{S}_2^{\mathcal{T}}(ww) = (K\alpha+1)\cdot\mathcal{D}_2^{\mathcal{T}}(ww) - K\alpha\cdot\left(\mathcal{S}_1^{\mathcal{T}}(w)\right)^2$$

$$= (K\alpha+1)\cdot\mathcal{D}_2^{\mathcal{T}}(ww) - K\alpha\cdot\lambda\cdot\mathcal{D}_2^{\mathcal{T}}(ww) = \mathcal{D}_2^{\mathcal{T}}(ww)\cdot(K\alpha\cdot(1-\lambda)+1).$$

Recall that $\mathcal{S}_1^{\mathcal{T}}(w) = \mathcal{D}_1^{\mathcal{T},\alpha}(w)$ and $\widetilde{\mathcal{S}}_1(w) = \widetilde{\mathcal{D}}_1(w)$, so that $\widetilde{\mathcal{S}}_1(w) = (1 \pm \xi')\mathcal{S}_1^{\mathcal{T}}(w)$. Moreover, $\widetilde{\mathcal{S}}_2(ww) = (K\alpha + 1)\widetilde{\mathcal{D}}_2(ww) - K\alpha \cdot (\widetilde{\mathcal{S}}_1(w))^2$. We provide an upper bound for $\widetilde{\mathcal{S}}_2(ww)$:

$$
\begin{aligned}
\widetilde{\mathcal{S}}_2(ww) &\leq (1 + \xi') \cdot (K\alpha + 1) \cdot \mathcal{D}_2^{\mathcal{T},\alpha}(ww) - (1 - \xi')^2 \cdot K\alpha \cdot (\mathcal{S}_1^{\mathcal{T}}(w))^2 \\
&\leq (1 + \xi') \cdot (K\alpha + 1) \cdot \mathcal{D}_2^{\mathcal{T},\alpha}(ww) - (1 - 2\xi') \cdot K\alpha \cdot (\mathcal{S}_1^{\mathcal{T}}(w))^2 \\
&= (1 + \xi') \cdot (K\alpha + 1) \cdot \mathcal{D}_2^{\mathcal{T},\alpha}(ww) - (1 - 2\xi') \cdot K\alpha \cdot \lambda \cdot \mathcal{D}_2^{\mathcal{T},\alpha}(ww) \\
&= \mathcal{D}_2^{\mathcal{T},\alpha}(ww) \cdot (K\alpha(1 - \lambda) + 1 + \xi'(K\alpha(1 + 2\lambda) + 1)) \\
&\leq \mathcal{D}_2^{\mathcal{T},\alpha}(ww) \cdot (K\alpha(1 - \lambda) + 1 + \xi'(3K\alpha + 1)) \\
&\leq \mathcal{D}_2^{\mathcal{T},\alpha}(ww) \cdot (K\alpha(1 - \lambda) + 1 + \xi) \\
&\leq \mathcal{D}_2^{\mathcal{T},\alpha}(ww) \cdot (K\alpha(1 - \lambda) + 1 + \xi \cdot (K\alpha(1 - \lambda) + 1)) \\
&= (1 + \xi) \cdot \mathcal{D}_2^{\mathcal{T},\alpha}(ww) \cdot (K\alpha(1 - \lambda) + 1) \\
&= (1 + \xi) \cdot \mathcal{S}_2^{\mathcal{T}}(ww).
\end{aligned}
$$

The other direction is analogous:

$$
\begin{aligned}
\widetilde{\mathcal{S}}_2(ww) &\geq (1 - \xi') \cdot (K\alpha + 1) \cdot \mathcal{D}_2^{\mathcal{T}}(ww) - (1 + \xi')^2 \cdot K\alpha \cdot (\mathcal{S}_1^{\mathcal{T}}(w))^2 \\
&\geq (1 - \xi') \cdot (K\alpha + 1) \cdot \mathcal{D}_2^{\mathcal{T},\alpha}(ww) - (1 + 2\xi' + (\xi')^2) \cdot K\alpha \cdot (\mathcal{S}_1^{\mathcal{T}}(w))^2 \\
&\geq (1 - \xi') \cdot (K\alpha + 1) \cdot \mathcal{D}_2^{\mathcal{T},\alpha}(ww) - (1 + 3\xi') \cdot K\alpha \cdot (\mathcal{S}_1^{\mathcal{T}}(w))^2 \\
&= (1 - \xi') \cdot (K\alpha + 1) \cdot \mathcal{D}_2^{\mathcal{T},\alpha}(ww) - (1 + 3\xi') \cdot K\alpha \cdot \lambda \cdot \mathcal{D}_2^{\mathcal{T},\alpha}(ww) \\
&= \mathcal{D}_2^{\mathcal{T},\alpha}(ww) \cdot (K\alpha(1 - \lambda) + 1 - \xi'(K\alpha(1 + 3\lambda) + 1)) \\
&\geq \mathcal{D}_2^{\mathcal{T},\alpha}(ww) \cdot (K\alpha(1 - \lambda) + 1 - \xi'(4K\alpha + 1)) \\
&\geq \mathcal{D}_2^{\mathcal{T},\alpha}(ww) \cdot (K\alpha(1 - \lambda) + 1 - \xi) \\
&\geq \mathcal{D}_2^{\mathcal{T},\alpha}(ww) \cdot (K\alpha(1 - \lambda) + 1 - \xi \cdot (K\alpha(1 - \lambda) + 1)) \\
&= (1 - \xi) \cdot \mathcal{D}_2^{\mathcal{T},\alpha}(ww) \cdot (K\alpha(1 - \lambda) + 1) \\
&= (1 - \xi) \cdot \mathcal{S}_2^{\mathcal{T}}(ww).
\end{aligned}
$$

### A.3 Proof of Lemma 5

Let $i^\star \in [n]$ be an integer such that $|v(i^\star)| = |v|_\infty$. We begin with the lower bound on $|v|_p$:

$$
|v|_p^p = \sum |v(i)|^p \geq |v(i^\star)|^p = |v|_\infty^p = (1 - \epsilon)^p \cdot |v|_1^p.
$$

We now move on to the upper bound on $|v|_p$. If $n = 1$, the upper bound is trivial, since all the $p$-norms of any given 1-dimensional vector are identical. We then assume $n \geq 2$. Then,

$$
\begin{aligned}
|v|_p^p = \sum |v(i)|^p = \sum \left(|v(i)| \cdot |v(i)|^{p-1}\right) &\leq \sum \left(|v(i)| \cdot |v|_\infty^{p-1}\right) \\
&= |v|_1 \cdot |v|_\infty^{p-1} = (1 - \epsilon)^{p-1} \cdot |v|_1^p.
\end{aligned}
$$

### A.4 Proof of Theorem 6

We have

$$
\rho_w \geq \frac{K \cdot \mathcal{S}_2^{\mathcal{T}}(ww) \cdot (1 - \xi)}{K \cdot \left(\mathcal{S}_1^{\mathcal{T}}(w) \cdot (1 + \xi)\right)^2} = \frac{|x_w|_2^2 \cdot (1 - \xi)}{|x_w|_1^2 (1 + \xi)^2} \geq \frac{(1 - \epsilon_w)^2 (1 - \xi)}{(1 + \xi)^2},
$$

where the last inequality follows from Lemma 5. The other direction is analogous:

$$
\rho_w \leq \frac{K \cdot \mathcal{S}_2^{\mathcal{T}}(ww) \cdot (1 + \xi)}{K \cdot \left(\mathcal{S}_1^{\mathcal{T}}(w) \cdot (1 - \xi)\right)^2} = \frac{|x_w|_2^2 \cdot (1 + \xi)}{|x_w|_1^2 (1 - \xi)^2} \leq \frac{(1 - \epsilon_w)(1 + \xi)}{(1 - \xi)^2}.
$$

## A.5 Proof of Lemma 7

The first claim follows directly from the lower bound on $\rho_w$ of Theorem 6.

As for the second claim, suppose that $\rho_w \geq \frac{1-\xi}{(1+\xi)^2}$. By Theorem 6, we have that $\frac{(1-\epsilon_w)(1+\xi)}{(1-\xi)^2} \geq \rho_w$. Thus,

$$\frac{(1-\epsilon_w)(1+\xi)}{(1-\xi)^2} \geq \frac{1-\xi}{(1+\xi)^2} \iff 1 - \epsilon_w \geq \left(\frac{1-\xi}{1+\xi}\right)^3 \iff \epsilon_w \leq 1 - \left(\frac{1-\xi}{1+\xi}\right)^3.$$

Now, $\frac{1-\xi}{1+\xi} \geq \frac{(1-\xi)-\xi}{(1+\xi)-\xi} = 1 - 2\xi$, and, by the union bound, $(1-2\xi)^3 \geq 1 - 6\xi$. Hence, $\epsilon_w \leq 6\xi$.

## A.6 Proof of Theorem 8

By rearranging the terms in the reduction of Theorem 2, we have

$$\frac{\mathcal{D}_2^{\mathcal{T}}(w_1 w_2)}{\mathcal{D}_1^{\mathcal{T}}(w_1) \cdot \mathcal{D}_1^{\mathcal{T}}(w_2)} = \frac{1}{K\alpha + 1}\left(K\alpha + \frac{\mathcal{S}_2^{\mathcal{T}}(w_1 w_2)}{\mathcal{S}_1^{\mathcal{T}}(w_1) \cdot \mathcal{S}_1^{\mathcal{T}}(w_1)}\right).$$

If $w_1$ and $w_2$ are co-dominated, that is, the case where they have their largest probability on the same topic, we have:

$$K\mathcal{S}_2^{\mathcal{T}}(w_1 w_2) = \langle x_{w_1}, x_{w_2}\rangle \geq \prod_{i \in \{1,2\}} |x_{w_i}|_\infty = \prod_{i \in \{1,2\}} ((1-\epsilon_{w_i})|x_{w_i}|_1) = \prod_{i \in \{1,2\}} \left((1-\epsilon_{w_i})K\mathcal{S}_1^{\mathcal{T}}(w_i)\right),$$

implying

$$\tau(w_1, w_2) \geq \frac{(1-\xi)}{(1+\xi)^2} \cdot \frac{\mathcal{D}_2^{\mathcal{T}}(w_1 w_2)}{\mathcal{D}_1^{\mathcal{T}}(w_1) \cdot \mathcal{D}_1^{\mathcal{T}}(w_2)} \geq \frac{1}{K\alpha + 1}(K\alpha + K(1-\epsilon_1)(1-\epsilon_2)).$$

On the other hand, if $w_1$ and $w_2$ are not co-dominated, we have

$$K \cdot \mathcal{S}_2^{\mathcal{T}}(w_1 w_2) \leq \left(K \cdot \mathcal{S}_1^{\mathcal{T}}(w_1)(1-\epsilon_{w_1})\right)\left(K \cdot \mathcal{S}_1^{\mathcal{T}}(w_2)\epsilon_{w_2}\right) + \left(K \cdot \mathcal{S}_1^{\mathcal{T}}(w_1)\epsilon_{w_1}\right)\left(K \cdot \mathcal{S}_1^{\mathcal{T}}(w_2)(1-\epsilon_{w_2})\right)$$
$$+ \left(K \cdot \mathcal{S}_1^{\mathcal{T}}(w_1)\epsilon_{w_1}\right)\left(K \cdot \mathcal{S}_1^{\mathcal{T}}(w_2)\epsilon_{w_2}\right)$$
$$\leq (K \cdot \mathcal{S}_1^{\mathcal{T}}(w_1)) \cdot (K \cdot \mathcal{S}_1^{\mathcal{T}}(w_2)) \cdot (\epsilon_{w_1} + \epsilon_{w_2} + \epsilon_{w_1}\epsilon_{w_2}),$$

implying

$$\tau(w_1, w_2) \leq \frac{(1+\xi)}{(1-\xi)^2} \cdot \frac{\mathcal{D}_2^{\mathcal{T}}(w_1 w_2)}{\mathcal{D}_1^{\mathcal{T}}(w_1) \cdot \mathcal{D}_1^{\mathcal{T}}(w_2)} \leq \frac{1}{K\alpha + 1}(K\alpha + K(\epsilon_{w_1} + \epsilon_{w_2} + \epsilon_{w_1}\epsilon_{w_2})).$$

## A.7 Proof of Corollary 9

It is enough to show that the minimum $\tau(w_1, w_2)$ on pairs of words $w_1, w_2$ that are co-dominated is larger than the maximum $\tau(w_1', w_2')$ on pairs of words $w_1', w_2'$ that are not co-dominated. Hence, by Theorem 8, it is sufficient to show that

$$\frac{(1-\xi)}{(1+\xi)^2} \cdot \frac{K\alpha + K(1-\epsilon)^2}{K\alpha + 1} > \frac{(1+\xi)}{(1-\xi)^2}\frac{K\alpha + K(2\epsilon + \epsilon^2)}{K\alpha + 1} \iff \left(\frac{1-\xi}{1+\xi}\right)^3 > \frac{\alpha + 2\epsilon + \epsilon^2}{\alpha + (1-\epsilon)^2}.$$

For the LHS, we have $\frac{1-\xi}{1+\xi} \geq \frac{(1-\xi)-\xi}{(1+\xi)-\xi} = 1 - 2\xi$, and $(1-2\xi)^3 \geq 1 - 6\xi$. The RHS is equivalent to $1 - \frac{(1-\epsilon)^2 - 2\epsilon - \epsilon^2}{\alpha + (1-\epsilon^2)} = 1 - \frac{1-4\epsilon}{\alpha + (1-\epsilon)^2} \leq 1 - \frac{1-4\epsilon}{\alpha+1}$. Then, it suffices for $\xi$ to satisfy $1 - 6\xi > 1 - \frac{1-4\epsilon}{\alpha+1} \iff \xi < \frac{1}{6}\frac{1-4\epsilon}{\alpha+1}$.

## A.8 Proof of Theorem 10

We analyze each step of Algorithm 1:

1. Consider the set $W = \left\{w \in \mathcal{V} \mid \widetilde{\mathcal{D}}_1(w) \geq \frac{p}{2K}\right\}$ of words of empirical frequency at least $\frac{p}{2K}$. By definition every anchor word has probability at least $\frac{p}{K}$; by Lemma 3(b), if $n \geq \left\lceil \frac{K}{p} \cdot \frac{9}{\delta^2} \ln m \right\rceil$, every anchor word $w$ satisfies $\widetilde{\mathcal{D}}_1(w) \geq (1-\delta)\mathcal{D}_1^{\mathcal{T},\alpha}(w) \geq (1-\delta)\frac{p}{K} \geq \frac{p}{2K}$, so every actual anchor words belong to $W$.

2. Applying Lemma 3(b) with $q = \left(\frac{p}{2K}\right)^2$ and $\xi = \frac{\delta}{4(K\alpha+1)}$, we can obtain $\widetilde{\mathcal{D}}_1(w)$ and $\widetilde{\mathcal{D}}_2(ww)$ within a $\left(1 \pm \frac{\delta}{4(K\alpha+1)}\right)$ multiplicative error for all words $w \in W$.

3. Theorem 4(b) immediately implies that the reduction of of Theorem 2 provides an estimate $\widetilde{\mathcal{S}}_2(ww)$ within a $(1 \pm \delta)$ multiplicative error from $\mathcal{S}_2^{\mathcal{T}}(ww)$, for any $w \in W$.

4. We now apply Lemma 7 to obtain the set $A$ of quasi-anchor words (that is, of words $w$ whose vector of probabilities $x_w$ has large $\ell_1$-weight, at least $p/2$, and such that at least $1 - \epsilon_w \geq 1 - 6\delta \geq 1 - \frac{6}{48} \geq \frac{7}{8}$ of their weight belongs to a single topic.)

5. Let $E$ be the maximal subset of $\binom{A}{2}$ such that $\widetilde{\mathcal{D}}_2(w_1 w_2) = (1 \pm \xi)\mathcal{D}_2^{\mathcal{T}}(w_1 w_2)$ for each $\{w_1, w_2\} \in E$. We prove that each co-dominated pair $\{w, w'\}$ is part of $E$. Observe that for any co-dominated pair $\{w, w'\}$, the reduction of Theorem 2 implies that $\mathcal{D}_2^{\mathcal{T},\alpha}(ww') \geq \frac{1}{K\alpha+1}\mathcal{S}_2^{\mathcal{T}}(ww') \geq \frac{(1-\epsilon)^2}{K\alpha+1}\mathcal{S}_1^{\mathcal{T}}(w)\mathcal{S}_1^{\mathcal{T}}(w') \geq \left(\frac{7}{8}\right)^2 \frac{p^2}{K(K\alpha+1)}$. Another application of Lemma 3(b) with $q = \left(\frac{7}{8}\right)^2 \frac{p^2}{K(K\alpha+1)}$ and $\xi = \delta$ ensures that $\widetilde{\mathcal{D}}_2(w, w') = (1 \pm \delta)\mathcal{D}_2^{\mathcal{T},\alpha}(ww')$. Hence, all co-dominated pairs belong to $E$.

6. At this point the algorithm has obtained $K$ pairwise non-codominated quasi-anchor words say $w_1, \ldots, w_K$. For each $i \in [K]$, the $i$th vector will be defined to be equal to $t_i(w) \leftarrow \frac{\widetilde{\mathcal{S}}_2(w_i w)}{\widetilde{\mathcal{S}}_1(w_i)}$, for each $w$ in the vocabulary. Recall that each $w_i$ is such that $|x_{w_i}|_\infty \geq (1 - 6\delta)|x_{w_i}|_1$; let $j$ be the topic such that $x_{w_i}(j) = |x_{w_i}|_\infty$. Then, $\frac{\widetilde{\mathcal{S}}_2(w_i w)}{\widetilde{\mathcal{S}}_1(w_i)} = x_w(j) \pm O(\delta)$. Thus, we can reconstruct all the probabilities of the topic that dominates $w_i$, to within an additive $O(\delta)$ error. Since no two $w_i$'s are codominated, and since there are $K$ distinct $w_i$'s, there will exist a bijection $\phi$ from $\{t_1, \ldots, t_K\}$ to $\mathcal{T}$ such that $|t_i - \phi(t_i)|_\infty \leq O(\delta)$.