[Reviews · NeurIPS 2018]

Reviewer 1



his paper presents a novel approach for reducing the latent Dirichlet allocation (LDA) to a single topic allocation (STA) for more efficient and accurate reconstruction of the underlying topics. I find the idea quite interesting, but I have the following concerns. First, this paper has many important parts missing and relies on other sources -- unpublished manuscript to show the equivalence of uniform LDA and STA, and "full version" for the general (p,t)-separable case. What would be this full version paper? As far as I know, NIPS conference papers should be mostly self-contained, except for some parts that rely on previous literature. While the appendix does include the unpublished manuscript, it is not required for the reviewers, and quite frankly this appendix is too lengthy and dense to review for accuracy. This paper would be better published as a longer, perhaps a journal, paper. Next, in the experiments, why do you use very different document lengths for Recover/STA vs. Gibbs sampling? I understand that Gibbs sampling needs longer documents. Then why not us the same lengths for Recover/STA? Would there be issues with computation time? In practice, you would expect the document lengths to be closer to 10 or 100, the lengths used for Gibbs sampling, rather than the extreme length of 2. Also, how was the parameter \rho set? Does the different settings of that parameter affect the results (i.e., loss and time)? It seems different settings would result in different numbers of anchor and quasi-anchor words. What are some anchor words found? As done in Arora paper, readers would probably like to see more details about the experiments and the results. Lastly, I am not convinced that this problem of topic reconstruction is well motivated. The point of LDA is not just discovering a set of topics for a corpus, but also discovering the topic distribution for each document, such that it can be used for downstream tasks, such as document retrieval or classification. Although topic reconstruction itself is important, given the other concerns I expressed above, I would be hesitant to accept this paper as part of the NIPS conference. Minor question: In line 167, distribution induced by STA should be S, not D? *** Update *** I appreciate the author response. I had overlooked the contributions of reconstruction of the topics, and I have gone back to previous relevant papers in NIPS and similar venues to confirm. That was my main issue, and thus I increased the score. I also understand the authors' point about the document length issue, while far from what would be used in practice.

Reviewer 2



This submission presents an approach to LDA topic reconstruction using word co-occurences. Exploiting a relationship between LDA and multinomial mixture models (detailed in a separate, included paper), the authors provide a new algorithm for learning (a broad class) of separable topic models. These results are tested on separable data and compared to Arora et al (2013)’s anchor words algorithm and a Gibbs sampler. I will disclaim that I am comparatively weak on the theoretical background for reviewing this paper. I come from a background quite familiar with LDA and various approaches to estimation. It may be worth noting for future readers that the abstract isn’t really clear that the goal here isn’t necessarily a practical algorithm as much as a theoretical contribution. The quality of the paper seems high, although again I’m not particularly well suited to judge many of the theoretical results. The paper considers two simulation settings, one that is favorable to the anchor words assumptions and one that is not. In the favorable setting performance is comparable, but STA is faster. In the unfavorable setting, STA is both faster and has much stronger performance. From a practical perspective, what is lacking is any sense of whether this works at all on real data. The anchor words algorithm actually works in real data and recovers semantically coherent topics. Does the proposed algorithm work for actual documents (particularly those longer than 2 words)? Does it scale well in the size of the vocabulary and when that vocabulary has a power-law distribution? It also seems that much of the results of experiment one turns on the runtime of STA being better, but that turns a lot on the implementation of the algorithms (which one was used for anchor words?). My reaction here may be due to the practical perspective I’m coming from, but it isn’t clear what the experiments would contribute for someone coming at this from a theory angle. The paper is remarkably clear. The authors do a nice job of signaling the work that each lemma and theorem will be doing to support the method. While the build-up of the approach is clear there seems to be a significant leap to the actual algorithm. It might help in Algorithm 1 to include references to the section where each step is described. I think it would be difficult to implement without constantly referring back to parts of the paper, but it isn’t always clear where to go (e.g. I struggled down the source of the threshold in step 2, because it wasn’t clear what what would happen at p >=2 where the empirical fraction would be above 1). Finally, it would really help if the appendix included further details of the experiment. For example, (1) timing comparisons to Gibbs sampling are always a bit misleading because there is no clear point of termination, so it can be made to look arbitrarily bad by just running it arbitrarily long, (2) it isn’t clear which recover algorithm is used (Recover-KL, Recover-L2 etc.), (3) what is used for topic reconstruction in Gibbs? Is it a single pass of the Gibbs sampler or an average over draws (which is going to be substantially more accurate), (4) there is a curious line which is “Gibbs needs longer documents in order to work so we use lengths \ell \in {10,100}, and we keep the number of total tokens (and thus documents) fixed.” How can you keep the number of total tokens and the number of documents fixed (I’m assuming relative to STA/Recover) if the length is different (because total tokens = number of documents * average length). The work seems quite original to me and represents a substantial move beyond other spectral method of moments recovery algorithms in the literature. It covers the interesting (p,t)-separable case. I’d be curious to know how it compares to the generalization of the anchor words method provided in Huang, Fu and sidiropoulos (“Anchor-Free Correlated Topic Modeling; Identifiability and Algorithm”) The significance of this papers substantially by community. Amongst the community of applied users of LDA, I suspect it will have little significance at the moment because the algorithm as stated throws away essentially all the data in the documents, it makes extremely strong assumptions (like the symmetric hyper parameter which we know from Wallach et al 2009 is important) and there is no evidence it works on real applications. It seems to represent a really new and interesting direction for theory but unfortunately I’m less well-suited to judge that. ====================== Response to Authors Reply: Thanks for the reply- this was really helpful for clarifying the details of the evaluations. I am sympathetic to the difficulties of doing the Gibbs runtime evaluations and I think being transparent about that in the text will help clarify to others why it is so difficult.

Reviewer 3



This paper presents a new algorithm for topic reconstruction named STA, which works under (p, 1)-separability. The proposed algorithm first follows an assumption that each document has only one topic (STA), so as to discover anchor words, and then uses the anchor words to reconstruct the topics of LDA. The proposed algorithm is reported to run faster than previous models ("Recover") in this line and Gibbs sampling. Quality: The general idea of the paper is interesting. Anchor words can be viewed as the most representative words of a topic. It is a smart idea to discover them with the single topic assumption. The paper gives strong theoretical derivations and analysis to the proposed algorithm, which definitely a plus to the paper quality. The main issue of the paper is on the experiments. Perhaps it would be a little bit harsh to argue the experiments of a theoretical paper like this one, but I would say the experiments are not convening. The issues are as follows: (1) How do the authors compare with Gibbs sampling, with different settings of the corpus? For Recover and STA, the document length is 2, while for Gibbs sampling, it is 10 to 100? (2) The running time comparison is also not rigorous. What are the reasons for using 200+1000 iterations? Does Gibbs sampling converge or not? Mallet is implemented in Java, so what is the implementation of STA and Recover? These factors can heavily affect the running time performance. More important, efficiency is one of the main claims of the paper. (3) It is a little bit ridiculous that l is set to 2. To me, it is hard to think a topic model can learn meaningful topics with documents with only 2 words. It is not a practical setting. (4) The authors should compare or discuss with the topic models using the spectral algorithm. Compared with Gibbs sampling, those models are efficient for large datasets as well. Clarity: To me, the paper is hard to follow. The paper uses a stack of texts, definitions, and theorems, which makes it hard to navigate in the paper. Originality and significance: I like the idea of this paper, which to me is novel. The significance of the paper mainly on the theoretical aspect. The proposed algorithm is theoretically guaranteed with complexity, but the experiments are not convening, which limits the usability and impact of the paper. --------------------------------------------------------------------------- Thanks for the reply. I think the idea is cool. But it needs improvements on the evaluation, like I've discussed in the previous review.